# On the Relationship between Suspended Sediment Concentration, Rainfall Variability and Groundwater: An Empirical and Probabilistic Analysis for the Andean Beni River, Bolivia (2003–2016)

**Irma Ayes Rivera [1],\*, Ana Claudia Callau Poduje [2], Jorge Molina-Carpio [3] , José Max Ayala [4], Elisa Armijos Cardenas [5], Raúl Espinoza-Villar [6], Jhan Carlo Espinoza [7] , Omar Gutierrez-Cori [8] and Naziano Filizola [1,9]**

[1]   Postgraduation program CLIAMB, Instituto Nacional de Pesquisas da Amazônia (INPA)—Universidade do Estado do Amazonas (UEA), Manaus CEP 69060-001, Brazil; nazianofilizola@ufam.edu.br
[2]   Institute of Horticultural Production Systems, Section of Vegetable Systems Modelling, Leibniz Universität Hannover, D-304019 Hannover, Germany; callau@gem.uni-hannover.de
[3]   Instituto de Hidráulica e Hidrología (IHH), Universidad Mayor de San Andrés, Casilla 699, Campus Universitario, Calle 30 Cota Cota, La Paz 15000, Bolivia; amolina@umsa.bo
[4]   Instituto Hondureño de Ciencias de la Tierra (IHCIT), Universidad Nacional Autónoma de Honduras (UNAH), Ciudad Universitaria, Boulevard Suyapa, Tegucigalpa 11101, Honduras; jmax_ayala@hotmail.com
[5]   Instituto Geofísico del Perú (IGP), Calle Badajoz 169, Urb. Mayorazgo IV etapa, Ate, Lima 15012, Peru; earmijos@igp.gob.pe
[6]   Universidad Nacional Agraria La Molina (UNALM), Ave. La Molina, S.N., Lima 15012, Peru; respinoza@lamolina.edu.pe
[7]   Univ. Grenoble Alpes, CNRS, IRD, Grenoble INP, IGE (UMR 5001), 38000 Grenoble, France; jhan-carlo.espinoza@ird.fr
[8]   Sorbonne Université, CNRS, Laboratoire de Météorologie Dynamique (LMD), Institut Pierre Simon Laplace (IPSL), Paris 75252, France; omar.gutierrez@lmd.jussieu.fr
[9]   Universidade Federal do Amazonas (UFAM), Ave. General Rodrigo Otávio, Jordão Ramos 6200, Campus Universitário, Coroado I, Manaus CEP 69077-000, Brazil
\*   Correspondence: irma.ayes@inpa.gov.br; Tel.: +55-9298-468-6783

**Abstract:** Fluvial sediment dynamics plays a key role in the Amazonian environment, with most of the sediments originating in the Andes. The Madeira River, the second largest tributary of the Amazon River, contributes up to 50% of its sediment discharge to the Atlantic Ocean, most of it provided by the Andean part of the Madeira basin, in particular the Beni River. In this study, we assessed the rainfall (R)-surface suspended sediment concentration (SSSC) and discharge (Q)-SSSC relationship at the Rurrenabaque station (200 m a.s.l.) in the Beni Andean piedmont (Bolivia). We started by showing how the R and Q relationship varies throughout the hydrological year (September to August), describing a counter-clockwise hysteresis, and went on to evaluate the R–SSSC and Q–SSSC relationships. Although no marked hysteresis is observed in the first case, a clockwise hysteresis is described in the second. In spite of this, the rating curve normally used (SSSC = $aQ^b$) shows a satisfactory $R^2 = 0.73$ ($p < 0.05$). With regard to water discharge components, a linear function relates the direct surface flow $Q_s$–SSSC, and a hysteresis is observed in the relationship between the base flow $Q_b$ and SSSC. A higher base flow index ($Q_b/Q$) is related to lower SSSC and vice versa. This article highlights the role of base flow on sediment dynamics and provides a method to analyze it through a seasonal empirical model combining the influence of both $Q_b$ and $Q_s$, which could be employed in other watersheds. A probabilistic method to examine the SSSC relationship with R and Q is also proposed.

**Keywords:** Andean Beni River; Bolivia; surface suspended sediment concentrations; rainfall; groundwater

---

## 1. Introduction

The erosion of the eastern flank of the tropical Andes is the principal provider of the sediment load observed in the Amazon Basin, with a major contribution from the Ucayali and Madeira Rivers [1–4], tributaries that originate in the southern Andes. The Madeira River alone contributes nearly 50% of the Amazon River sediment load, most of it provided by the Beni River, a tributary that rises in the Bolivian Andes, where sediment yields up to 18,000 t·km$^{-2}$·year$^{-1}$ have been estimated [5–8]. In fact, many geomorphological, biochemical and ecological features of the Amazon plain are linked to the magnitude and variability of water and sediment supplied from the Andes [9].

However, there is no agreement regarding the main factors that control erosion in the Bolivian Andes. Although for Dunne et al. [10] the main factor was vegetation cover, for Filizola et al. [7] it was rainfall and runoff, for Pepin et al. [11] it was climate variability, and Aalto et al. [12] found that topographic steepness and lithology appear to have the most active effects. By contrast, Safra et al. [13] suggested that climate and lithology do not appear to exert a primary control over the erosion rates in the Bolivian Andes. It is important to note that due to the non-linear relationship between the different factors that control erosion, it is often challenging to identify a dominant driving factor. Moreover, some of the disagreement arises from the variety of data and methodologies applied. In terms of relationships, Guyot et al. [14] found that water discharge and solid fluxes were related, but relationships between rainfall and discharge with sediment load have not been further assessed in this Andean sub-basin. Recent studies [15,16] have determined that groundwater is a significant contributor to the water discharge in many Amazon headwater sub-basins and could influence sediment concentrations as well.

The suspended sediment concentration (SSC), averaged at the river cross-section, is typically evaluated by their water discharge (Q) relationship. The most common empirical relationship between suspended sediment and water discharge is the power function SSC = aQ$^b$. Such an empirical relationship has several control factors embedded in the a and b coefficients. However, if hysteresis patterns in the Q–SSC relationship are found [17–20], this power function cannot be appropriately used. In the Amazon plain, more specifically at the station of Óbidos, a clockwise hysteresis between Q–SSC is found as a result of the different hydrological cycles from the main tributaries (Solimões, Negro and Madeira) [21]. Further developments on those empirical relationships are listed in Vercruysse et al. [22] including data mining techniques like fuzzy logic, artificial neuronal networks and principal components analysis. Copula functions were tested in Slovenian rivers, to estimate event-based SSC [23] and in the Weine River in China to evaluate the variations in the runoff-sediment relationship [24,25]. On the other hand, more straightforward and direct empirical models can be established by using the rainfall in the equations, with the additional advantage that rainfall data is more widely available [26]. The empirical Q–SSC relationship has also been enhanced by considering the surface and groundwater contribution separately, as observed in Senegal, in Mexico and Nepal [27–29]. In Senegal, the authors found that suspended sediment concentration is diluted by the subsurface and groundwater discharge [27]. Similar findings were found for Nepal, further related to the monsoon rainfall seasonality [29].

As climate change and human perturbations may have severe impacts on the hydrological cycle of the Upper Madeira sub-basin [30], including potential implications on the sediment yield and dynamics [31] of the Beni River sub-basin, we need to improve our knowledge of the rainfall-discharge-SSC relationship in this region, and also to generate estimation tools for assessing those impacts. This knowledge is further needed as there are proposals for big hydropower dams for this Andean-Amazon sub-basin near its outlet [32,33]. Accordingly, this study aims to (1) evaluate

the relationship of rainfall and water discharge components (i.e., direct and base flow) with surface suspended sediment concentration (SSSC) at the Rurrenabaque (RU) gauging station (at the Beni Andean sub-basin outlet) between 2003 and 2016 and (2) assess empirical and probabilistic methods to estimate monthly SSSC. Please note that we evaluated SSSC relationships only, as SSSC is directly recorded at Rurrenabaque, while SSC needs to be calculated with a power-rating curve [3], thus adding further uncertainties to the calculations.

## 2. Material and Methods

### 2.1. Study Area

The Upper Beni sub-basin is located to the east of the Andean Cordillera in Bolivia, between 14° and 18° S and 66° and 69° W. It extends over an altitude range of 200–6450 m a.s.l. and an area of 70,000 km$^2$, which represents 25% of the Beni sub-basin and 6.3% of Bolivia's territory (Figure 1a). McQuarrie [34] divided the sub-basin into four tectono-structural zones: (1) the Altiplano, a low-relief plateau in Quaternary rocks; (2) the Eastern Andean Cordillera, a thrust belt that deforms lower Paleozoic rocks; (3) the Inter-Andean zone, with a structure similar to that of the Eastern Cordillera, but with younger rocks and deformation at higher structural levels and (4) the Sub-Andes, consisting of faulted folds within Tertiary foreland sedimentary rocks.

The interaction between the complex topography and large-scale atmospheric circulation (e.g., South American Monsoon System, the South American Low-Level Jet, Southern winds intrusions, etc.), creates a complex spatial rainfall distribution, with annual precipitation ranging from under 300 to over 4500 mm·year$^{-1}$ [35–37]. For the 2003–2016 period at RU, the mean annual rainfall was estimated at 1230 mm by the satellite-based precipitation product CHIRPS.v2 (Climate Hazard Group Infrared Precipitation with Stations in its second version) [38], with more than 50% (770 mm) of the annual rainfall occurring during the austral summer (December–March, DJFM; Figure 1b) [35].

Climate and geology give origin to very contrasting landscapes: from semi-arid, highly erodible sub-basins incised in the Altiplano (La Paz River), to densely vegetated hyper-humid sub-basins on the Eastern Cordillera (Yungas valleys) and the Alto-Beni foothills in the Sub-Andes region [39]. Those factors also enable significant groundwater recharge in nearly 70% of the sub-basin, estimated at 100–300 mm·year$^{-1}$ [40] (Figure 1a), inferring that groundwater supports a significant part of the total streamflow, as described by [15].

Erosion at Angosto del Bala (former RU gauging station) and RU were estimated by Guyot et al. [6] and Vauchel et al. [3] in 3140 and 2745 t·km$^2$·year$^{-1}$ over two different periods, 1984–1989 and 2002–2011, respectively. Suspended sediment loads of nearly 212 and 192 Mt·year$^{-1}$ and mean annual water discharge (Q) of 1990 and 2014 m$^3$·s$^{-1}$ were estimated for the same locations and periods. Erosion rates estimated by Guyot et al. [6] at Angosto del Bala and three upstream stations are presented in Figure 1a. The largest erosion rate was estimated at 18,250 t·km$^2$·s$^{-1}$ for the La Paz River sub-basin at the Cajetillas station to the southwest of the Andean Beni River sub-basin. Suspended sediment load at RU represents 55% of the Beni River sediment load estimated at the Cachuela Esperanza outlet gauging station for the 2002–2011 period [3]. Regarding particle size distribution, suspended material seems to be dominated by fine sands and silts, with a clay fraction that exceeds 10% [39].

The annual evolution of the water discharge at RU shows a behavior similar to the CHIRPS' estimated rainfall for the study period, with a peak from January that extends to March. SSSC begins to increase in mid-September with the beginning of the rains, with a peak between January and February. Notably, as water discharge at RU during March is almost as high as in February (4000 m$^3$·s$^{-1}$), SSSC decreases from a mean value of 3100 mg·L$^{-1}$ in February to 1600 mg·L$^{-1}$ in March (Figure 1c,d).

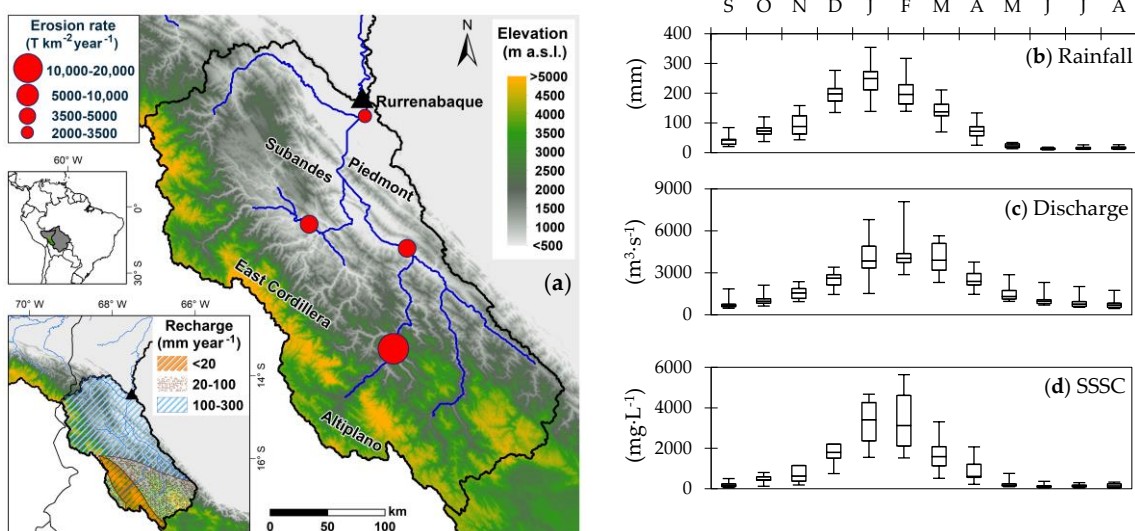

**Figure 1.** (**a**) Andean Beni sub-basin at the Rurrenabaque outlet gauge station (RU). Annual cycle (2003–2016) for (**b**) basin mean rainfall (R) at RU; (**c**) water discharge (Q) and (**d**) surface suspended sediment concentration (SSSC). Sources: for (**a**) Elevation data from Shuttle Radar Topography Mission (STRM) [41], sediment yields from Guyot et al. [6], and groundwater recharge from Bundesanstalt für Geowissenschaften und Rohstoffe (BGR) [40], (**b**) CHIRPS.v2 [38], (**c**) and (**d**) Hydrogéochimie du Bassin Amazonien (HYBAM), [42]. In (**c**) the 12,230 $m^3 \cdot s^{-1}$ measured during the extreme rainfall event in February 2014 is not plotted for visual purpose of the y-axis.

## 2.2. Data and Methodology

Mean Andean Beni sub-basin rainfall was estimated from the satellite-based precipitation product CHIRPS.v2 [38] with a 0.05° spatial resolution. This satellite dataset is freely available at the Climate Hazards Center website [43] and has been previously validated with an observed dataset for the Amazon Basin by Espinoza et al. [44] and for the Northern Altiplano by Sátge et al. [45]. Water discharge and instant surface suspended sediment concentration (SSSC) measured at RU for a common period were available at the HYBAM Observatory (HO) website [42]. SSSC measurements were interpolated to generate daily time-series through the Hydraccess program [46] and averaged to the monthly scale. Hydraccess is freely accessible at the HO website and offers several hydro-climatological data analysis tools. The measurement procedure and data quality can be read at Vauchel et al. [3]. We evaluated the time-series for the common period from March 2002 to October 2016, analyzing the temporal variability on a monthly time-scale, considering the hydrological year from September to December (year$_{t-1}$) to January to August (year$_{t0}$).

We analyzed possible relationships through linear Pearson correlation (r) and ranked Kendall coefficient (τ). After observing a high agreement between the time-series, we evaluated the power function based on the water discharge (Table 1). We also conducted separate evaluations of surface flow ($Q_s$) and base flow ($Q_b$; Table 1). The discharge components were separated at daily time step through the recursive digital filter, and then averaged to the monthly scale. The recursive filter uses a frequency analysis separating the $Q_s$ with the following equation [47]:

$$Q_{s,t} = \propto Q_{s,\,t-1} + \frac{(1+\propto)}{2}(Q_t - Q_{t-1}).$$

(1)

Here, $Q_{s,t}$ is the filtered surface discharge at the time step t; $Q_{s,t-1}$ is the filtered surface discharge at the t–1 time step; $\propto$ is the filter parameter and $Q_t$ is the total discharge at t and $Q_{t-1}$ is the total discharge at t–1. Hence the base flow is $Q_b = Q_t - Q_s$ and the base flow index (BFI) is the relationship between the base flow and the total discharge ($Q_b/Q$). $Q_s$ and $Q_b$ separation was achieved using the free Web-based Hydrograph Analysis Tool (WHAT) from Purdue University [48]. The $\propto$ filter parameter

ranges from 0.90 to 0.95. It was set to 0.93 in this study using the master recession curve (MRC) [47,49]. The MRC represents the summary of all the hydrograph's recession fragments (i.e., period of no or reduced recharge but with groundwater drainage, in this case from May to August), and relates these fragments through concatenation so that a main curve can be established with $MRC = Q_o e^{-\left(\frac{t}{k}\right)}$, $\propto = e^{-\left(\frac{1}{k}\right)}$ and with k as a recession constant.

We subsequently evaluated the multivariate distributions (i.e., rainfall, total water discharge and SSSC) using the Copula function so that we could model the joint cumulative distribution between the variables at a monthly time-scale. This function can model the dependence of uniformly transformed observations described as pseudo-observations [50,51]. For a bivariate case, the link between a Copula, denoted as $C(u_1,u_2)$, and bivariate distribution is provided by Sklar's theorem, with the Equation (5) presented in Table 1 [50]. In this equation, $F(x_1,x_2)$ is the joint cumulative distribution function with the continuous marginal distribution functions (i.e., each univariate probability distribution) of the random variables: $F_1(x_1)$ and $F_2(x_2)$.

To set up a Copula model the appropriate family must be selected through the Akaike information criterion (AIC) and the corresponding parameter(s) estimated for each family by statistical inference (maximum likelihood) [52,53]. Since each Copula depends on the type of dependence between the variables, we verified it through Kendall's τ. To model dependence among time-series using Copula, each time-series must be independent and identically distributed. The temporal dependence among the variables must therefore be eliminated beforehand. Several authors have applied Copula to model this type of variables based on the normalized time-series [54] or based on the standardized residuals resulting from time-series models applied to each variable separately [55,56]. In our work the original time-series were standardized, and when a temporal dependence was found after the Ljung-Box and Box-Pierce tests [57], an autoregressive-moving-average (ARIMA) model was applied to remove the temporal dependence, resulting in uncorrelated time-series of residuals. The ARIMA model takes the following form [58]:

$$y_t = \mu + \varnothing_1\big(y_{t-1} - \mu\big) + \varepsilon_t. \tag{2}$$

Here, $y_t$ is the value at the time step t; μ and $\varnothing_1$ are fitted parameters of the model and $\varepsilon_t$ is the uncorrelated residual of the time series been analyzed (also known as the noise).

We conducted a frequency analysis based on the time-series to define the best distribution for modeling these margins. Several probability distributions were included in the analysis, which are described by different numbers of parameters. Additionally, we considered several methods of estimating the parameters (method of moments, maximum pseudo-likelihood and L-Moments).

Finally, we conducted SSSC estimation based on Q and R as follows: (1), for each pair of variable Q–SSSC and R–SSSC we conducted a simulation of 20,000 uniform random variables (0,1) by using each Copula; (2), the mean value of all 20,000 possible pseudo-SSSC was selected to estimate SSSC and (3) the two mean pseudo-SSSC time-series were transformed from the copula space (0,1) to the real space as standardized time-series, which were then converted to SSSC. Note that other parameters (median and quartiles) from each simulation can also be used to estimate SSSC.

All the estimations for univariate and copula analyses presented here were performed using R [59]. The parameters of the marginal distributions were estimated by the L-moments method using the Lmomco package [60], and method of moments and maximum likelihood with the Fitdistrplus package [61]. The Copula [62–65] and VineCopula [66] packages were used for the parameter estimation of the copula models and the synthesis of random pairs of dependent variables.

**Table 1.** Functions and dependences evaluated. Here $Q_s$ stands for the surface component, $Q_b$ for the baseflow and $\varepsilon$ for the error.

| Dependence | Type of Function Analyzed | Equation |
|---|---|---|
| SSSC = f(Q) | SSSC = $aQ^b + \varepsilon$ | Equation (3) |
| | SSSC = $aQ_s + bQ_b + \varepsilon$ | Equation (4) |
| SSSC = f(R) SSSC = f(Q) Bivariate Copula | $F(x_1, x_2) = C[F_1(x_1), F_2(x_2)]$ | Equation (5) |

The statistical criteria used for evaluating the performance of the different methods were the percent bias (PBIAS), Nash–Sutcliffe efficiency (NSE), coefficient of determination ($R^2$), Kendall's $\tau$ and the maximum under- and over-estimation range (Table 2). PBIAS measures the tendency of simulated data to be larger or smaller than the observed counterpart. The NSE is a normalized statistic for determining the relative magnitude of the residual variance compared with the measured data variance [67], indicating the agreement between observed versus simulated data. The coefficient of determination ($R^2$) describes the proportion of the variance in measured data explained by the model. It ranges from 0 to 1, with values closer to 1 indicating less error variance.

**Table 2.** Statistical criteria used for evaluation. Performance ratings based on Moriasi et al. [68]. Here xobs(i) refers to observed data at month i, with $\overline{xobs}$ as the mean and ymod(i) for model estimation with $\overline{ymod}$ as the mean. For Kendall's Tau: $P_n$ number of concordant pairs, with n as the number of pairs; two pairs $(X_i, Y_i),(X_j, Y_j)$ being concordant if $(X_i - X_j) \cdot (Y_i - Y_j) > 0$.

| | |
|---|---|
| Percent Bias Equation (6) | $PBIAS = \left[ \frac{\sum_{i=1}^{n}(xobs(i) - ymod(i))}{\sum_{i=1}^{n}(xobs(i))} \right] \times 100$ |
| Nash-Sutcliffe Efficiency Equation (7) | $NSE = 1 - \left[ \frac{\sum_{i=1}^{n}(xobs(i) - ymod(i))^2}{\sum_{i=1}^{n}\left(xobs(i) - \overline{xobs}\right)^2} \right]$ |
| Linear Coefficient of determination Equation (8) | $R^2 = \frac{\left[\sum_{i}^{n}\left(xobs(i) - \overline{xobs}\right)(ymod(i) - \overline{ymod})\right]^2}{\sum_{i}^{n}\left(xobs(i) - \overline{xobs}\right)^2 \sum_{i}^{n}\left(ymod(i) - \overline{ymod}\right)^2}$ |
| Kendall's Tau Equation (9) | $\tau_n = \frac{4}{n(n-1)}P_n - 1$ |

| Performance Rating | PBIAS | NSE | $R^2$ |
|---|---|---|---|
| Very good | $< \pm 10$ | $0.75 < NSE \leq 1.00$ | |
| Good | $\pm 10 \leq PBIAS < \pm 15$ | $0.65 < NSE \leq 0.75$ | $\geq 0.60$ |
| Satisfactory | $\pm 15 \leq PBIAS < \pm 25$ | $0.50 < NSE \leq 0.65$ | |
| Unsatisfactory | $\geq \pm 25$ | $\leq 0.50$ | |

Additionally, a cross-validation approach was adopted. First, functions were generated by removing one hydrological year in each step. Then estimates of the removed year were calculated by the generated functions. Finally, the statistical metrics used for the performance evaluation were employed for both the removed and non-removed years. In the case of the bivariate Copula functions, family and parameters stability were also compared.

## 3. Results and Discussions

### 3.1. Multivariate Time-Series

The relationship between the monthly time-series was assessed through Kendall's correlation coefficients. All were significant at 99% of confidence (Table 3). R–Q had a higher correlation after a one-month lag with $\tau = 0.70$ ($\tau = 0.57$ without lag). However, correlation $\tau = 0.65$ with the SSSC is the same for both Q and a one-month lag (Q-Lag) series. The one-month lag and the discharge peak that lasts until March indicate the progressive drainage of water storage in the sub-basin. The absence of

lag between the rainfall and SSSC responds to the short time concentration, of about three days [69] related to the direct flow.

**Table 3.** Kendall's τ for rainfall (R), discharge (Q), discharge after one-month lag (Q-Lag) and surface suspended sediment concentration (SSSC) at monthly scale (2003–2016).

| Kendall's τ | R | Q | Q-Lag | SSSC |
|---|---|---|---|---|
| R | - | 0.57 | 0.70 | 0.72 |
| Q | 0.57 | - | 0.61 | 0.65 |
| Q-Lag | 0.70 | 0.61 | - | 0.65 |
| SSSC | 0.72 | 0.65 | 0.65 | - |

A R–Q counter-clockwise hysteresis can be observed, with different relationships established for the start of the rainfall in September to its peak in January period, and for the February to August period (Figure 2a). This reflects the increase of water contribution from the whole basin from the start of the rainfall, with subsequent rainfall and surface contribution cessation. The R–SSSC relationship describes a potential function for the same periods analyzed between R–Q. Though an evident hysteresis was not observed, this relationship was lost in February and modified for the rest of the year (parameter a changes from 0.23 to 2.13; Figure 2b). The R–SSSC relationship shows a differentiated spatial distribution into the December–April wet season (Figure 3). While positive significant correlations were observed during the rainfall peak in January in 90% of the sub-basin ($r > 0.5$, $p < 0.05$), correlation was lost during the discharge peak in February followed by an increase from March in the south-eastern part of the sub-basin that lasts until April ($r > 0.8$, $p < 0.05$). The most erodible region (the one with the highest sediment yield) could therefore not be identified by studying spatio-temporal relationships between CHIRPS data and SSSC values throughout the year.

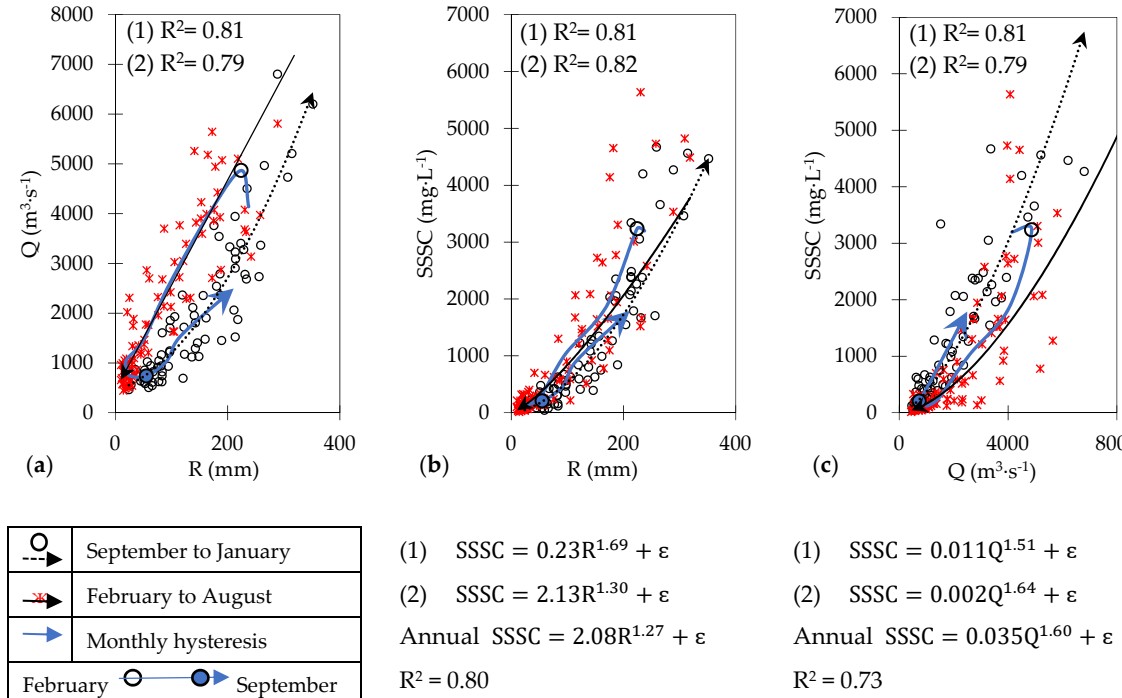

**Figure 2.** Monthly time-series relationships 2003–2016: (**a**) mean Andean Beni sub-basin R in mm vs. Q at RU in $m^3 \cdot s^{-1}$; (**b**) as (**a**) between R–SSSC at RU in $mg \cdot L^{-1}$ and (**c**) as (**a**) between Q–SSSC. First $R^2$ representing September–January and second $R^2$ for February–August periods.

The Q–SSSC relationship displays a clockwise hysteresis with a power function ($R^2 = 0.73$, $p < 0.05$) over the annual scale with fitted parameters (a and b) that differ when evaluating the R–Q hysteresis

period (Figure 2c). As observed by Andermann et al. [29] in the Himalayas, the fact that water discharge displays a clockwise hysteresis within the SSSC suggests that material supply varies between seasons. The authors also observed a counter-clockwise hysteresis between R–Q, which was related to the transient storage and release of water in the basin from the basement aquifer, which in turn could control the sediment concentration transported by the river because of the dilution differential.

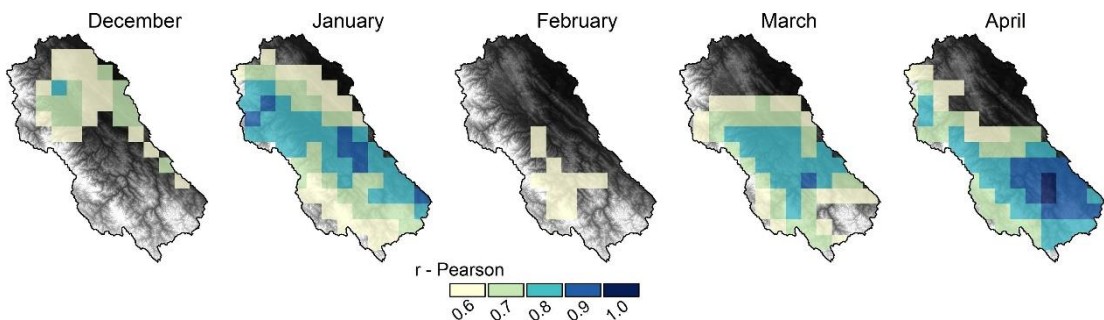

**Figure 3.** CHIRPS' rainfall spatial r-Pearson correlation ($\alpha = 0.05$) with monthly SSSC measured at RU in the Andean Beni River from December to April.

### 3.2. The Relationship of Water Discharge Components ($Q_s$ and $Q_b$) with SSSC

After separating the direct flow ($Q_s$) and base flow ($Q_b$) from the total water discharge, a hysteresis pattern emerges from the relationship between temporal $Q_b$–SSSC series, and a linear relation between $Q_s$–SSSC (Figure 4a,b). The hysteresis could be distinguished in the following three periods: the first from the start of the rainfall in September until its peak in January (SONDJ) when concentration increased from the previous dry season (SSSC = $0.90Q_s$, Figure 4b); during the second period, R and $Q_s$ decreased, which generated the start of SSSC decline from February until April SSSC = 0.76 $Q_s$, (Figure 4b); this reduction was further pronounced for the following months (May–August dry season), which was the third hysteresis period until the next rainy season. We observed this hysteresis throughout all the fourteen years analyzed, including the heavy rainfall event of the austral summer of 2013–2014 [70], with an extended influence in 2015 (Figure 4a). Regarding the BFI, the relationship with SSSC shows an exponential decay ($R^2 = 0.76$), with SSSC less than 1000 mg·L$^{-1}$ for a ratio larger than 0.60. A very dominant base flow (BFI > 0.90) was associated with a very low mean SSSC value of 140 mg·L$^{-1}$ and a coefficient of variability (CV) of 0.81 (Figure 4c). Conversely, the diminution of base flow was accompanied by a varied increase of sediment concentration. The SSSC shows a high variability (CV of 0.99) at the start of the rainy season, which decreased to 0.72 after its February peak. The $Q_s$ contribution thus presents a linear relationship for the annual cycle within the suspended sediments ($R^2 = 0.66$). However, if we consider the hysteresis periods and their relationships with the discharge components, we see the following fitted parameters:

$$\text{SSSC}_{\text{annual}} = \left(0.90Q_{s-\text{SONDJ}} + 160\right) + \left(0.75Q_{s-\text{FMA}} + 346\right) + \left(0.034Q_{b-\text{MJJA}}^{1.20}\right) + \varepsilon. \tag{10}$$

Notably, the total discharge Q vs. concentrations in the previous section, combined the solid matter mobilized by the direct flow $Q_s$ and by the effect of groundwater $Q_b$. Groundwater generated the dilution of sediment concentrations and increased the river transport capacity. This suggests that the suspended matter in this sub-basin was not dependent on the magnitude of water discharge but on the transport capacity that exceeds the material mobilized by the direct flow. As described by Santini et al. [71] on the Peruvian Andean Amazon Rivers (Ucayali and Marañon), when the fine suspended fraction is dominant, the wash load depends on different factors such as the matter availability, rainfall and sediment entrainment processes that occur on the hillslopes, which can act alone or in combination.

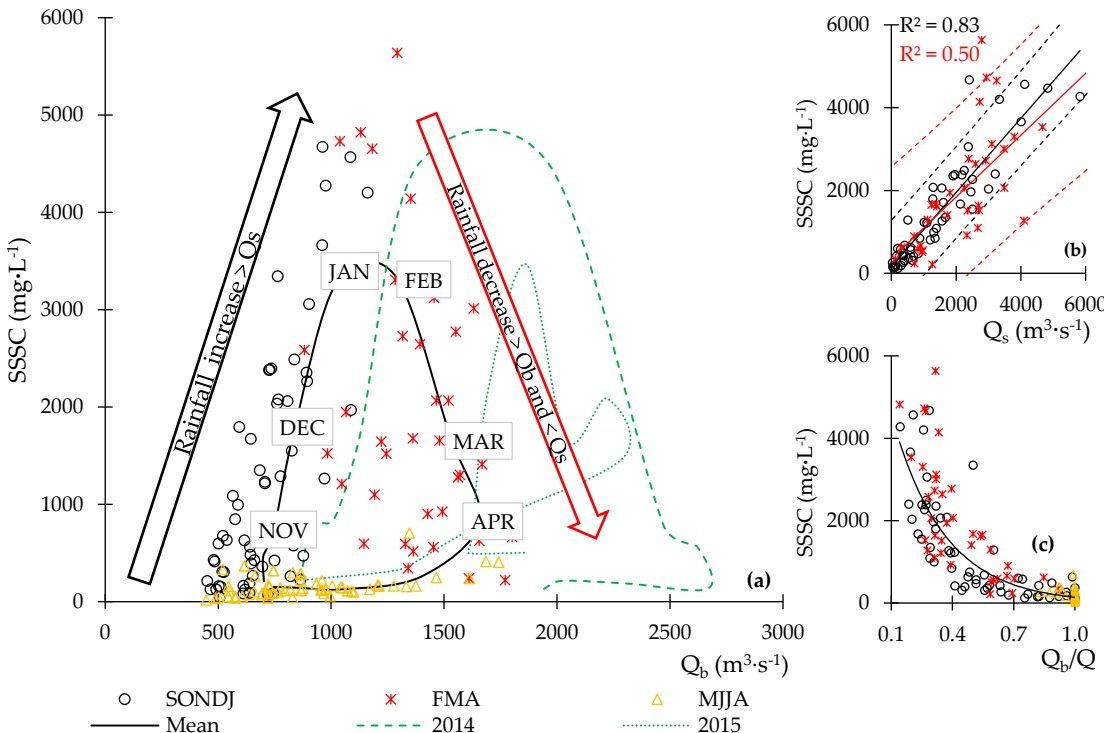

**Figure 4.** (**a**) Base flow ($Q_b$) vs. SSSC at RU in the Andean Beni River (2003–2016) at monthly scale. (**b**) Surface discharge ($Q_s$) vs. SSSC; here black (red) line for the linear relationship and dashed black (red) lines for the 95% confidence interval for September to January—SONDJ (February to March, FMA). (**c**) Base Flow Index (BFI) vs. SSSC.

Andermann et al. [29] observed in Himalayan' rivers that annual groundwater fluctuations are associated with the rainfall monsoon seasonality. Due to this, the authors concluded that the total discharge could not be used for estimating sediment concentration unlike direct surface flow. Likewise, our results were in accordance with the hysteresis found between the water discharge and the conductivity (related to dissolved solid material) described at Rurrenabaque by Moquet et al. [72]. The latter authors suggested that during the rainy season (November–May), the highest concentrations were found in response to surface runoff mobilization, while during the dry period, the variability would be mainly controlled by the dilution associated with groundwater inputs.

### 3.3. Bivariate Copula Functions

To find a joint Copula function between the marginal distributions, we first verified the autocorrelation from the three standardized time-series. There was no significant autocorrelation for the R–SSSC within the Ljung-Box and Box-Pierce test $p < 0.39$ and $p < 0.29$, respectively. However, since water discharge is a highly autocorrelated variable, as was observed ($p < 0.0001$), we used an autoregressive-moving-average (ARIMA) model to removed it and use the uncorrelated residual time-series ($p < 0.21$, from now on $Q_{re}$). As a result, the Kendall's correlation coefficients for the dependence between the variables were $\tau = 0.35$ ($p < 0.0001$) for the R–SSSC relationship and $\tau = 0.34$ ($p < 0.0001$) for $Q_{re}$–SSSC.

After evaluating different distribution functions, the marginal distributions for the variables were: Pearson 3 (NSE = 0.99; BIAS = 0.004) for the rainfall, generalized extreme value (NSE = 0.99; BIAS = 0.005) for the $Q_{re,}$ and Weibull 3 for the SSSC (NSE = 0.99; BIAS = 0.006). The joint functions were then determined, and in both cases a Gaussian Copula was selected after a minimum AIC value, –52.64 and –46.72, for R–SSSC and $Q_{re}$–SSSC, respectively. For the R–SSSC function the Gaussian parameter = 0.54 was determined and for the $Q_{re}$–SSSC, 0.52. With the defined Copula functions, we made the two simulations of 20,000 pseudo Copula anomalies. From each simulation, we estimated

the mean value and transformed it to the standardized SSSC values, which were later converted to SSSC (estimated SSSC in Section 3.4).

### 3.4. SSSC Estimations Assessments Performance

The SSSC estimations based on the functions above result in the following four time-series (Figure 5). These are presented within their cumulative distribution function (CDF) to be able to compare not only the annual cycle but also to represent the observed SSSC parameters for the 2003–2016 period. When we considered only the power function as it was normally used, it is notable that in years with two sediment peaks, the first one was not represented as well as the second one (e.g., 2002–2003 and 2006–2007). We believe this was related to the change of relationship observed within the R–Q hysteresis periods (Figure 2a). Indeed, the first peak occurred in January while the second occurred in March. The recession period in the sedimentological curve was well represented by most of the years; however, it presented a delay during 2010 and 2011 and had no relationship to the extreme rainfall event of 2013–2014 (Figure 5a). The CDF shows that low SSSC denoted by the percentile 25 (P-25) was well represented (+14%) by the power function. However, from the percentile 75 (P-75) the parameter was below 20% and below 30% for the percentile 90 (P-90).

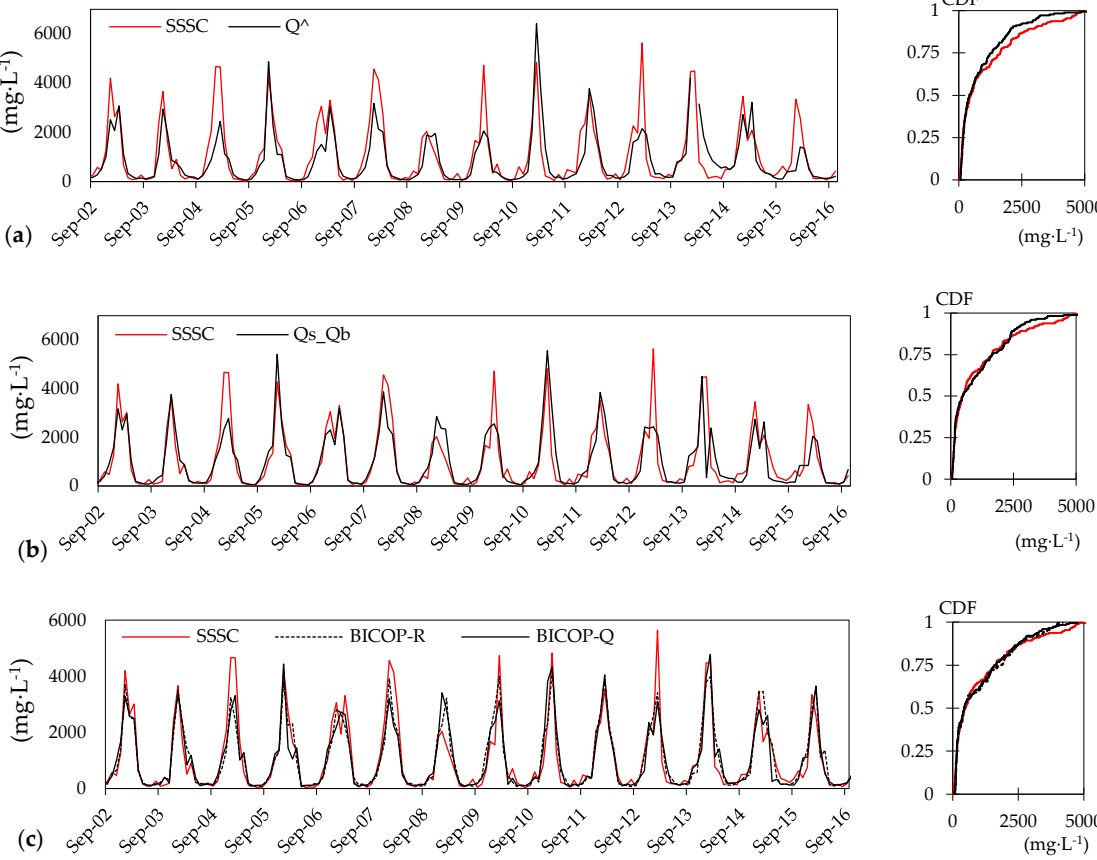

**Figure 5.** Observed SSSC (red line) at RU from September 2003 to September 2016 with the four models: (**a**) here Qˆ stands for the power functions; (**b**) Qs_Qb for the seasonal discharge components function and (**c**) BICOP-R and BICOP-Q for the two bivariate Copula functions, while CDF stands for the cumulative distribution function.

The seasonal function between the water discharge components and SSSC described better SSSC peaks and the rising period, as observed in 2003–2004 and 2007–2008 (Figure 5b). This suggests that the rising limb and first peak in January were better represented by the direct discharge and not by the total discharge. Nevertheless, unlike the power function, it could not estimate the highest peaks

(e.g., 2004–2005, 2009–2010 and 2012–2013). The former was observed in the CDF, in which the P-90 was underestimated by 16%. However, the CDF shows an improvement for the range between P-50 and P-75 in +1% and could represent the absence of or low direct discharge (between May and August) overestimating the P-25 by +3%.

In contrast with the above analysis, when we evaluated the bivariate Copula functions, the interannual and CDF improved, although in years with two peaks in some cases, the peaks were lost (e.g., 2006–2007). It is noteworthy that for the two Copula simulations, there were no major differences when considering the rainfall or the water discharge as independent variables, so even though it is a probabilistic model, it can be a useful statistical technique for estimating SSSC based only on one of the variables. Nevertheless, high SSSC was underestimated, with the P-90 below 10%, a P-50 underestimate of nearly 10%, while in both cases the P-25 (+3%; Figure 5c) was well represented.

Analyzing the above time-series generated in terms of their modeling statistics, more information from the classical discharge power function to the bivariate Copula functions led to a reduction in the PBIAS and an increase in the $R^2$ and NSE, varying the range of under- and over-estimation (Table 4). What is of interest is that even the Copula functions had very good performance ratings over the complete time-series estimation, but they underestimated the two highest SSSC months (i.e., January and February). The observed mean peak in 2005 and 2008, 4660 and 4350 mg·L$^{-1}$, respectively were poorly estimated by the bivariate Copula functions, at nearly –36% and –31%, while the lowest observed mean peak in 2009 of 1780 mg·L$^{-1}$ was overestimated by more than +68%. Indeed, none of the functions were able to estimate the highest sediment peak during 2008. It is important to point out that the study period was too short, and that of the 176 months available of SSSC, only 10 months had concentrations larger than 4000 mg·L$^{-1}$, which made it impossible for all the functions to represent these concentrations. Between January and February 2014 alone, instant measurements were larger than 5300 mg·L$^{-1}$, while in February 2013 a 15,100 mg·L$^{-1}$ was reported. Though values larger than 4000 mg·L$^{-1}$ might be considered as outliers, they could represent flash floods that can occur over this piedmont station, affecting the daily time-series interpolations and monthly aggregation [3].

**Table 4.** Statistical criteria obtained for the different functions (2003–2016). In all cases Kendall's τ was significant at the 99% level.

| Function | Kendall's τ (–) | Linear R$^2$ (–) | Max Underestimation/Max Overestimation (mg·L$^{-1}$) | PBIAS (%) | NSE (–) |
|---|---|---|---|---|---|
| $SSSC = 0.003Q^{1.60} + \varepsilon$ | | | | | |
| | 0.66 | 0.65 | −2400/4500 | 17.7 | 0.58 |
| $SSSC = \left(0.90Q_{s-SONDJ} + 160\right) + (0.75Q_{s-FMA} + 346) + \left(0.034Q_{b-MJJA}^{1.20}\right) + \varepsilon$ | | | | | |
| | 0.72 | 0.72 | −2130/4140 | 5.0 | 0.71 |
| $SSSC = $ Gaussian Bivariate Copula $+ \varepsilon$ | | | | | |
| SSSC_1 = (R − SSSC) | 0.75 | 0.85 | −1600/2500 | 4.6 | 0.84 |
| SSSC_1 = (Q − SSSC) | 0.75 | 0.83 | −1400/2600 | 5.2 | 0.83 |

Results of the cross-validation indicate that both the 2005 drought and 2014 flood events were better represented by the seasonal function than the potential (Table 5). Which in turn had a similar performance as the one estimated by the bivariate Copula functions. Additionally, through this procedure we could observe that the joint behavior of Q–SSSC was adequately represented by the Gaussian Copula in 13 out of the 14 steps. Moreover the estimated parameters were comparably ranging from 0.49 to 0.55 (0.52 for the all period, Section 3.3), indicating an analogous correlation for all years. Conversely, R–SSSC showed to have different joint behavior for the analyzed years as different types of Copula were selected (Gaussian in six cases, Gumbel in five and Tawn in three). This suggests a difficulty of the bivariate Copula to represent the rainfall variability and its dispersion in the joint function with the SSSC.

**Table 5.** Cross-validation statistical criteria obtained for the different functions. The intervals represent the range of the metric for the estimated years in each validation step, and 2005 and 2014 as example for extreme events.

| Function | Kendall's $\tau$ (–) | Linear $R^2$ (–) | Max Underestimation/Max Overestimation (mg·L$^{-1}$) | PBIAS (%) | NSE (–) |
|---|---|---|---|---|---|
| SSSC $= aQ^b + \varepsilon$ | (0.38,0.85) | (0.34,0.95) | −3620/3435 | (−77,49) | (0.14,0.93) |
| 2005 | 0.85 | 0.86 | −370/3130 | 46 | 0.86 |
| 2014 | 0.38 | 0.61 | −2900/430 | −77 | 0.61 |
| SSSC $= aQ_s + bQ_b + \varepsilon$ | (0.64,0.97) | (0.44,0.97) | −2435/3340 | (−73,29) | (0.35,0.94) |
| 2005 | 0.91 | 0.93 | −440/2380 | 29 | 0.68 |
| 2014 | 0.83 | 0.85 | −1575/214 | −27 | 0.79 |
| SSSC $=$ Bivariate Copula $+ \varepsilon$ | | | | | |
| SSSC_1 $= (R - SSSC)$ | (0.53,0.91) | (0.78,0.97) | −1495/2640 | (−41,28) | (0.43,0.97) |
| 2005 | 0.91 | 0.97 | −380/2050 | 28 | 0.77 |
| 2014 | 0.53 | 0.90 | −1000/610 | −7 | 0.90 |
| SSSC_1 $= (Q - SSSC)$ | (0.52,0.85) | (0.64,0.97) | −1630/2,410 | (−48,31) | (0.4,0.84) |
| 2005 | 0.85 | 0.90 | −640/1910 | 22 | 0.75 |
| 2014 | 0.64 | 0.78 | −1500/920 | −10 | 0.79 |

## 4. Conclusions and Perspectives

Two approaches were used to estimate the surface suspended sediment concentration (SSSC) for the Andean Beni sub-basin in Bolivia based on its rainfall and water discharge relationships (2003–2016). First, we evaluated the power function regularly used to estimate SSSC based on water discharge, followed by the assessment of the SSSC relationship between the surface and base flow contribution. The former approach was compared with SSSC estimates based on the probabilistic Copula function through statistical criteria (i.e., PBIAS, $R^2$, NSE, Kendall's $\tau$ and under/overestimation range). We also compared the results based on the model's ability to achieve satisfactory annual behavior.

Our results confirmed the dual role rainfall had on the suspended sediment dynamics, first by promoting solid material detachment and moving it to the river channel by the surface flow and second, by recharging the groundwater, which in turn promoted the dilution of the sediment concentration. Our results provided a further understanding of the relationships between these variables in this Andean Amazon sub-basin. Nevertheless, there are other factors and characteristics within the rainfall and water discharge that need to be further evaluated (including total duration of rainfall events, intensity and antecedent rainfall conditions including soil wetness) at other spatial and temporal scales.

The main results of this research were:

1. Although the power rating curve presented a satisfactory $R^2 = 0.73$, it was insufficient to estimate SSSC, because it did not account for scatters along the regression curve, caused by events with high SSSC and the hysteresis between the variables (rainfall vs. water discharge and base flow vs. SSSC).

2. The evaluation of the SSSC differentiating the contribution of direct surface and base flows from total water discharge allowed us to see the role these components had in the sediment dynamics in this sub-basin. Thus SSSC was estimated through a sum of seasonal functions based on surface and base flow contributions. Considering other floodplain areas and aquifers, mainly at the sub-basin's headwaters in the Amazon, future sediment dynamic research could take into account the potential role of base flow in the suspended sediment concentration.

3. By considering the time-series' marginal distributions in a bivariate Copula function, we reduced the PBIAS significantly to less than 6% and achieved a very good NSE of 0.83. Furthermore, the annual cycle could be reproduced satisfactorily. However, it is not only the stability of the time-series, which must be continuously evaluated to search for changes in the distributions' parameters that can modify the Copula function; it is also necessary to consider that as a probabilistic technique it still fail to establish the physical understanding that relates the variables.

Moreover, a further evaluation could consider using a multivariate Copula to estimate SSSC based on both (rainfall and discharge) in the same joint function.

Finally, we acknowledge that the selection of the most suitable model depends not only on the availability and quality of the input data and statistical technique, but also on the purpose of the model user. However, it is useful to combine different methods when trying to capture the underlying SSSC drivers and their dynamics.

**Author Contributions:** Conceptualization, methodology, and software developed by I.A.R., A.C.C.P. and J.M.A. Validation and formal analysis conducted by I.A.R, A.C.C.P., J.M.A., J.M.-C., J.C.E. and E.A.C. Research conducted by I.A.R., J.M.-C., J.C.E., A.C.C.P., E.A.C. and O.G.-C. Original draft prepared by I.A.R., A.C.C.P. and J.M.-C. Reviewing and editing carried out by I.A.R., A.C.C.P., J.M.A., J.M.-C., J.C.E., R.E.-V., E.A.C., O.G.-C. and N.F. Visualization designed by I.A.R., O.G.-C., and E.A.C.

**Funding:** The first author was supported by a Doctoral Scholarship from the Coordination for Improvement of Higher Education Personnel (CAPES) of the Brazilian Ministry of Education. J.M.-C. was supported by the Universidad Mayor de San Andres (UMSA) within the framework provided by the Hydrogéochimie du Bassin Amazonien (HYBAM) program and PHYBAAM (Processus Hydrologiques des Bassins Andins Amazoniens) project. J.C.E. was partially supported by the French AMANECER-MOPGA project funded by ANR and IRD (ref. ANR-18-MPGA-0008). The Brazilian Amazon State financed article publication through Fundação de Amparo à Pesquisa do Estado do Amazonas (FAPEAM), 005/2019.

**Acknowledgments:** We would like to thank the Bolivian SENAMHI and the HYBAM Observatory of the Institut de Recherche pour le Développement (IRD) for providing the hydrological and sediment data, and the Climate Hazards Group for generating the satellite-rainfall based product used in this study. The first author thanks the Peruvian Geophysics Institute for the internship received and the Hydraulic and Hydrology Institute of Bolivia's San Andrés Mayor University. We would also like to thank Alessandro Michiles and Paula Fonseca at UEA from Brazil and two anonymous reviewers for their comments and recommendations that helped to improve the manuscript.

**Conflicts of Interest:** The authors declare no conflict of interest. The founding sponsors had no role in the design, analysis, and interpretation of data, in writing the manuscript, or in the decision to publish the results.

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
