# Peer review of "On the Relationship between Suspended Sediment Concentration, Rainfall Variability and Groundwater: An Empirical and Probabilistic Analysis for the Andean Beni River, Bolivia (2003–2016)"

_water, doi:10.3390/w11122497_

Round 1

Reviewer 1 Report

This study provides a valuable contribution on the relationship between suspended sediment concentration, rainfall variability and groundwater. I believe it is a worthwhile contribution to the journal, but I have some comments to further improve the paper.

Major Comments

Abstract: The abstract lists major findings from the study, but fails to place these findings to a broader context. What are the implications and what are the relevance to other watersheds? The abstract should be concluded with 1-2 sentences that highlight the contributions and implications of the work, as opposed to facts from this work.

Model validation: The analysis doesn’t have any details on model validation. To the minimum, the authors may keep some portion of the 2015 or 2016 data as testing data and use the rest to train their model. This should provide a more objective evaluation of the model performance.

Terminology: The authors use SSC in some cases but SSSC (surface SSC) in other cases, which need to be clarified and better distinguished.

Baseflow separation: The author used 0.93 as alpha, but how does this choice affect the results and conclusions? I don’t expect much difference, but it would be good to have some sensitivity analysis to confirm that is the case.

Line 116: What is CHIRPS.v2? It should be explained with some details.

Line 151: Add some details for Hydraccess program.

Line 193: ARIMA should be described in more details, including its equation.

Editorial Comments

Line 163: Equation should be numbered

Line 203: convert à converted

Line 310: Because of

Line 322: high autocorrelated -> highly autocorrelated

Figure 1: Panels (b)-(d) should include tick marks and units for the y-axis labels.

Figure 5: Note the non-English characters in the plot

Author Response

We would like to thank the reviewer for his/her helpful comments. Please find below our responses.

Reviewer 2 Report

On the relationship between suspended sediment concentration, rainfall variability and groundwater: an empirical and probabilistic analysis for the Andean Beni River, Bolivia (2003-2016)

This paper presents an interesting analysis related to suspended sediment concentration dynamics and the discharge data methods used to predict them, a topic of persistent focus due to its significance in monitoring streams and rivers.

The paper is very detailed and comprehensive in its use of statistical models to analyse and evaluate prediction of SSC.  This leads to some sound conclusions in identifying the relationship baseflow and surface flow have on SSC values, and the seasonal pattern of these relationships.

However, there is some room for improvement. Some of the topics introduced in the introduction seem a bit disconnected to the focus of the paper and could do with some better linking or clarifying the context e.g. in the intro SSC-Q hysteresis seems to be introduced in the context of rising and falling hydrograph limbs, but the focus is more on monthly/seasonal hysteresis patterns (I think I interpreted that correctly).  Also, the terminology seems to jump around a bit, and I’m not sure why it begins introducing SSC then starts talking about SSSC. It isn’t clear if this is actually different. There is some room for improving the flow of the article to make it read smother. Also, I find he colour/shape formats used in the graphs to be quite confusing so some thought could be given to colour selection and layout to improve the visual readability of these figures.

Specific Comments

Title

Not sure the phrasing of the start of the title is the best choice (e.g. starting with a preposition) but probably just personal preference

Introduction

Line 79: This statement about counter-clockwise (or anti-clockwise) hysteresis seems to be back to front. Do you mean total ‘cumulative total suspended sediment for the entire rising limb? Counter-clockwise SSC-Q hysteresis occurs when SSC on the rising is lower than the falling, clockwise SSC-Q hysteresis occurs when SSC is higher on the rising and lower on the falling. See: 

Lloyd, C. E. M., Freer, J. E., Johnes, P. J., & Collins, A. L. (2016). Using hysteresis analysis of high-resolution water quality monitoring data, including uncertainty, to infer controls on nutrient and sediment transfer in catchments. Science of the Total Environment, 543, 388-404. Lawler, D. M., Petts, G. E., Foster, I. D., & Harper, S. (2006). Turbidity dynamics during spring storm events in an urban headwater river system: The Upper Tame, West Midlands, UK. Science of the Total Environment, 360(1-3), 109-126.

Also just need to make sure it is clear which type of hysteresis you are referring to, e.g. Q-SSC hysteresis.

Line 89 -91: Since this literature provides the context to the wider paper, it probably needs a bit more added to provide information/context on the findings of some of the studies you cited.

Line 95: ‘Indeed’ doesn’t make a lot of sense here, consider a different word or phrase.

Line 98 The introduction began with SSC literature, with a brief reference to separating surface and ground water which refers to the discharge. Then a new term SSSC is introduced without much explanation. How is SSSC different to SSC exactly, and how is it devised? I gather from the referenced paper it is modelled from SSC, Is there any extra uncertainty in using a derived/modelled value for these analysis?

Materials and Methods

Line 125: Are these sediment yields estimated as part of this study or from prior work? Is there a map of sediment yield?

Line 167:  Could this be phrased in a clearer way by replacing ‘effected’ with ‘achieved’ ?

Results and Discussions

Is it possible to be more consistent with specifying the relationships between each set of variables being referred? It seems to jump a bit between ‘Q and SSSC’ and ‘Q-SSSC’, or ‘R and Q’ and ‘R-Q’.

Line 239 Need to be clearer by specifying the hysteresis relationship of R and Q, i.e. are you referring to the hysteresis between R and Q, or with R and SSSC, and Q and SSSC?  

Line 261: Figure 2: This figure and caption could benefit from some reformatting for clarity. A key/legend is also useful (and quicker) for readers instead of writing everything in the caption. I do suggest in addition to the colours, use a shape component in case people are viewing in black and white. Also, the arrows are a bit confusing, especially the double black line which is hard to see. Perhaps there is a way to make that clearer using colour variation (or linking the line colour to a similar hue of the dots).  The label for the y and x axis should really sit on the axis it labelling, ideally on the ‘outside’ of the marked values, as per usually graphing format.

Maybe I didn’t understand, but in the introduction, you introduced counter-clockwise hysteresis in the context of SSC-Q hysteresis in a rising and falling limb of a hydrographs. However, it seems you are talking about seasonal SSC-Q hysteresis patterns. I think this needs more clarification in the introduction.

Line 273: Should this be Figure 4a, and 4b?

Line 296 Figure 4: The line colours used here confuse the image quite a bit because they use the same colours as the points. This implies they are related, but really, they’re displaying quite separate information. Consider changing the 2014 and 2015 line to an independent colour.

Line 310:  replace ‘o’ with ‘of’

Line 314: repeated word ‘authors’

Line 364: Figure 5: Looks like the axis labels haven’t come out right?

Conclusions

Nice conclusion

Author Response

(The authors gave the same response as above.)
